# Silver Nanoparticles Induce Neutrophil Extracellular Traps Via Activation of PAD and Neutrophil Elastase

**DOI:** 10.3390/biom11020317

**Published:** 2021-02-19

**Authors:** HanGoo Kang, Jinwon Seo, Eun-Jeong Yang, In-Hong Choi

**Affiliations:** 1Department of Microbiology, Institute for Immunology and Immunological Diseases, Yonsei University College of Medicine, Yonsei-ro 50-1, Seodaemun-gu, Seoul 03722, Korea; hgkang@yuhs.ac (H.K.); jinwon_seo@kolon.com (J.S.); yej5644@gmail.com (E.-J.Y.); 2Brain Korea 21 PLUS Project for Medical Science, Yonsei University College of Medicine, Yonsei-ro 50-1, Seodaemun-gu, Seoul 03722, Korea

**Keywords:** silver nanoparticle, neutrophil extracellular trap, peptidyl arginine deiminase

## Abstract

Silver nanoparticles (AgNPs) are widely used in various fields because of their antimicrobial properties. However, many studies have reported that AgNPs can be harmful to both microorganisms and humans. Reactive oxygen species (ROS) are a key factor of cytotoxicity of AgNPs in mammalian cells and an important factor in the immune reaction of neutrophils. The immune reactions of neutrophils include the expulsion of webs of DNA surrounded by histones and granular proteins. These webs of DNA are termed neutrophil extracellular traps (NETs). NETs allow neutrophils to catch and destroy pathogens in extracellular spaces. In this study, we investigated how AgNPs stimulate neutrophils, specifically focusing on NETs. Freshly isolated human neutrophils were treated with 5 or 100 nm AgNPs. The 5 nm AgNPs induced NET formation, but the 100 nm AgNPs did not. Subsequently, we investigated the mechanism of AgNP-induced NETs using known inhibitors related to NET formation. AgNP-induced NETs were dependent on ROS, peptidyl arginine deiminase, and neutrophil elastase. The result in this study indicates that treatment of 5 nm AgNPs induce NET formation through histone citrullination by peptidyl arginine deiminase and histone cleavage by neutrophil elastase.

## 1. Introduction

Silver nanoparticles (AgNPs) are one of the most widely used engineered nanoparticles (NPs) in a variety of fields because of their strong antimicrobial properties [1,2]. AgNPs are used in many industrial settings, including food preservation, cosmetics, and in biomedical applications such as wound dressings, to prevent contamination from microorganisms. As the use of AgNP-related product increases, the chance to expose to AgNPs are increasing [3]. For this reason, it is important to study the effects of NPs on the human immune system. The immune system defends the body against foreign antigens, which include NPs. Titanium dioxide NPs increased toll-like receptor (TLR)-10 at the mRNA level in human macrophage cell lines and silicon dioxide NPs reportedly increased TLR-9 and differentiated naive macrophages into inflammatory M1 macrophages [4]. In our previous study, inflammasome formation and interleukin (IL)-1β release was observed when human blood monocytes were exposed to AgNPs [5]. In case of AgNPs, reactive oxygen species (ROS) are a key factor of cytotoxicity in mammalian cells and an important factor in the immune reaction of neutrophils. AgNPs induced oxidative burst in neutrophils [6,7], cytokine release [8], and apoptosis [7,8,9].

Neutrophils are the most abundant immune cells in human blood (approximately 60% in leukocytes) and are the first cell type to arrive at a site of infection. When neutrophils encounter pathogens, they generally activate nicotinamide adenine dinucleotide phosphate (NADPH) oxidase to generate a large amount of ROS [10]. ROS are a key factor in the immune responses of neutrophils, such as phagocytosis, granule release, and neutrophil extracellular traps (NETs) Silver nanoparticles (AgNPs) are one of the most widely used engineered nanoparticles (NPs) in a variety of fields because of their strong antimicrobial properties [11,12]. NETs are webs of expelled DNA surrounded by histones and antimicrobial proteins such as neutrophil elastase (NE), LL-37, and myeloperoxidase (MPO) [13]. NETs entangle pathogens with sticky DNA and destroy them using a granular protein. However, excessive NET formation can cause several autoimmune diseases by acting as a source of autoantigen [14,15] or by promoting tissue damage [16,17].

Several NPs are known to induce NETs. The scanning electron microscope images of NETs induced by cationic lipid NPs were found [18] and polystyrene and diamond NPs induce NETs through ROS production consequence of lysosome leakage [19]. AgNPs also reported to have the ability to induce NETs but their mechanisms were unknown [20]. 

NETs are induced by a variety of stimuli, and each induced NET has different characteristics [21]. Mia Philipson and Paul Kubes have classified NETs into three types [22]. The first type includes suicidal NETs, which use the rupture of the plasma membrane for DNA release [12,23]. These NETs require ROS and peptidyl arginine deiminase type IV (PAD4)-mediated citrullination of histones. PAD4 converts positively charged arginine in histone tails, which helps to bind negatively charged DNA to uncharged citrulline and eventually induces chromatin decondensation [24]. Suicidal NET formation takes 3–4 h. The most common type of NET formation is induced by phorbol 12-myristate 13-acetate (PMA), IL-8, and some bacteria. 

The second type of NET includes vital NETs, which use vesicles to release DNA [25]. Vital NETs are ROS-independent and do not require rupture of the plasma membrane. Vital NET formation occurs relatively early (30–60 min) and is induced by *Staphylococcus aureus* and *Entamoeba histolytica*. The third type of NET includes mitochondrial NETs, which release mitochondrial DNA [19]. This type of NET has been rarely reported. 

However, this classification of NETs is insufficient to explain their induction by various stimuli. *Leishmania* triggers ROS-dependent NET formation in 1 h, and nicotine induces NETs depending on PAD4 but independent of ROS [26,27]. Some NET formation is induced through cell death due to necroptosis and autophagy [28,29]. Although NETs were discovered in 2004 [13], their classification remains in progress. 

In this study, we investigated the effects of AgNPs on human neutrophils, specifically NETs. Due to the diversity of NETs, we focused on detecting the mechanism and characteristics of AgNP-induced NETs rather than by classifying the type of NETs. We found that 5 nm AgNPs induce NETs and it is dependent on histone citrullination by PAD and histone degradation by NE.

## 2. Materials and Methods

### 2.1. Reagents

Phorbol 12-myristate 13-acetate (PMA), N-acetylcysteine (NAC), dibenziodolium chloride (DPI), trolox, L-ascorbic acid, and cytochalasin D were purchased from Sigma-Aldrich (St. Louis, MO, USA). MitoTEMPO, Go6976, PD98059, and BML-257 were purchased from Enzo Life Sciences (Farmingdale, NY, USA). BB-Cl-amidine, and 4-aminobenzoic acid hydrazide (ABAH) were purchased from Cayman Chemical (Ann Arbor, MI, USA). Chloroquine was purchased from Invivogen (San Diego, CA, USA). GW311616A was purchased from Adipogen Life Sciences (San Diego, CA, USA). 1,2-Bis (o-aminophenoxy)ethane-*N*,*N*,*N*′,*N*′-tetraacetic acid (acetoxymethyl ester) (BAPTA-AM) was purchased from Thermo Fisher Scientific (Waltham, MA, USA).

### 2.2. Characterization of AgNPs

AgNPs suspended in water were provided by I&C (5 nm; Seoul, Korea) or were purchased from ABC Nanotech (100 nm; Daejeon, Korea). AgNPs were spherical and coated with polyvinylpyrrolidone (PVP). Diameters were determined by transmission electron microscopy (TEM) (JEM-1011; JEOL, Tokyo, Japan) (6.2 ± 5.3 nm for 5 nm NPs and 83 ± 28.9 nm for 100 nm NPs, Appendix A). Distribution of AgNPs in 1% fetal bovine serum (FBS)-RPMI 1640 medium was investigated using dynamic light scattering (DLS; Malvern Instruments, Novato, CA, USA) (5.5 nm for 5 nm NPs and 83.4 nm for 100 nm NPs by number, Appendix A, middle panel). The AgNPs used in this study were prepared by steric stabilization using high-molecular-weight PVP; therefore, they did not easily aggregate. Endotoxin contamination was tested by pyrogen recombinant factor C assay (Cambrex Bioscience, Walkersville, MD, USA). Endotoxin was found to be negative (<0.01 unit/mL). Before the treatment of cells with AgNPs, polymyxin B was added to the culture medium as an endotoxin neutralizer to remove the potential contamination of NPs. Neutrophils express the endotoxin receptor toll-like receptor 4, which can activate neutrophils and induce NETs [23].

### 2.3. Neutrophil Isolation

Neutrophils were isolated from the venous blood of healthy volunteers using ethylenediaminetetraacetic acid (EDTA) anti-coagulated vacutainer tubes (BD Biosciences, San Jose, CA, USA), and a negative selection kit (EasySep™ Direct human neutrophil isolation kit; STEMCELL Technologies (Vancouver, British Columbia, Canada)) was then used according to the manufacturer’s protocol. Contamination of erythrocytes was removed with a red blood cell lysis buffer (Sigma-Aldrich). The purity of isolated neutrophils was more than 95% (Appendix A). Isolated human neutrophils were cultured in 1% FBS-RPMI 1640 medium at 37 °C in a humidified 5% CO_2_ incubator. 

### 2.4. Assessment of NET Formation

This experiment referenced white et al. [30] and modified. Since we don’t know which type of NETs induced by AgNPs, neutrophils treated with AgNPs for 4 h, which time to take suicidal NET formation. Freshly isolated human neutrophils (1 × 10^5^ cells/175 μL) were seeded in a 96-well plate and treated with AgNPs or PMA (25 μL) for 4 h at 37 °C and 5% CO_2_. Subsequently, 15 μL of DNase was added (1 unit/mL; Promega, Madison, WI, USA), and the samples were incubated for 15 min at room temperature. The plate was centrifuged for 10 min at 1800× *g*. Post-centrifugation, 100 μL of the supernatant was transferred to a black 96-well plate, and dsDNA was measured using a Quant-iT™ PicoGreen™ dsDNA Assay Kit (Thermo Fisher Scientific) according to the manufacturer’s instructions. Fluorescence was measured using a Victor X4 multilabel plate reader (PerkinElmer, Waltham, MA, USA).

To identify the characteristics of the NETs, extracellular DNA was observed by confocal microscopy. Freshly isolated neutrophils (5 × 10^4^ cells/175 μL) were seeded on 0.001% poly-l-lysine (Sigma-Aldrich)-coated eight-well chamber slides (Thermo Fisher Scientific). Neutrophils were treated with AgNPs or PMA (25 μL) for 4 h at 37 °C and 5% CO_2_. After fixation with 4% paraformaldehyde, samples were stained with SYTOX Green (1 μM, 15 min) (Thermo Fisher Scientific) and observed using confocal microscopy (FV1000, Olympus, Tokyo, Japan).

### 2.5. ROS Production

Isolated human neutrophils were pre-stained with 2 μM of 5-(and-6)-chloromethyl-2′,7′-dichlorodihydrofluorescein diacetate, acetyl ester (CM-H2DCFDA, Thermo Fisher Scientific) for 30 min in conical tubes and washed with PBS. Then, the cells (5 × 10^5^ cells/875 μL) were seeded on a 24-well plate and treated with AgNPs (125 μL) for 1.5 h at 37 °C and 5% CO_2_. For ROS measurement, we chose the mid time point to because ROS generation are preceding process of NET formation. Neutrophils were detached with a scrapper and transferred to a 5 mL polystyrene round-bottom tube and washed with ice-cold PBS for flow cytometry, which was performed using FACSCelesta (BD Biosciences) and analyzed using FlowJo software (FlowJo, Ashland, OR, USA).

### 2.6. Histone Citrullination

After the stimulation of neutrophils, cells were fixed with 4% paraformaldehyde. The fixed cells were permeabilized with 70% ethanol and blocked with 1% bovine serum albumin (BSA) and 10% FBS. Neutrophils were stained with anti-citrullinated histone H3 antibody (Abcam, Cambridge, MA, USA) as the primary antibody and with Alexa Fluor^®^ 594 conjugated goat anti-rabbit antibody (Abcam) as the secondary antibody. Flow cytometry was performed using FACSCelesta and analyzed using FlowJo software.

### 2.7. Histone Degradation

Neutrophils were treated with AgNPs or PMA for 2 or 3 h, and whole histones were isolated using a Histone Extraction Kit (Abcam). Since we wanted to see process inside neutrophils before NET formation, we chose time point for 2 and 3 h. If histone degradation experiment performed in 4 h, dead neutrophils affect total histone quantification. Proteins were blotted to a nitrocellulose membrane and blocked with 5% BSA in tris-buffered saline with Tween 20. The membrane was stained with histone H2B antibody (Cell Signaling Technology, Danvers, MA, USA) overnight at 4 °C with peroxidase-labeled anti-mouse antibody (KPL, Gaithersburg, MD, USA).

### 2.8. DNA Purification and PCR

The concept of this experiment referenced Yousefi et al. [31] and modified. Freshly isolated human neutrophils were stimulated with 5 nm AgNPs (1.6 μg/mL) or PMA (25 μL) for 4 h at 37 °C and 5% CO_2_. DNA purification from stimulated neutrophils was performed using a QIAamp^®^ DNA Blood mini kit (QIAGEN, Hulsterweg, The Netherlands). Isolated DNA was amplified with nuclear genes (*GAPDH* and *FAS*) and mitochondrial genes (*MT-ND1* and *MT-CYB*) by PCR. The primer sequences are summarized in Appendix A. The initial denaturation step was performed at 94 °C for 1 min, followed by 29 cycles of denaturation at 94 °C for 30 s, annealing at 51.4 °C for 50 s, and extension at 72 °C for 50 s. PCR products were separated by electrophoresis using 3% agarose gel.

## 3. Results

### 3.1. Induction of NET Formation by 5 nm AgNPs

To investigate whether 5 nm AgNPs could induce formation of NETs, extracellular DNA was measured (Figure 1A). The extracellular DNA gradually increased in a dose-dependent manner until the 5 nm AgNPs reached 1.6 μg/mL. The 100 nm AgNPs did not induce extracellular DNA up to 2 μg/mL. To confirm whether this extracellular DNA comprised NETs, the confocal images of DNA treated with 5 nm AgNPs or PMA, a representative NETs inducer, were acquired (Figure 1B). Human neutrophils were treated with 1.6 μg/mL 5 nm AgNPs or 5 ng/mL PMA for 4 h and stained with SYTOX Green for DNA staining. The cloudy form of NETs was observed in 5 nm AgNP-treated neutrophils, similar to that in PMA-treated neutrophils. Since NET formation is accompanied by membrane rupture, 5 nm AgNPs concentration used in this study was determined lethal (1.6 μg/mL) and sublethal (1.2 μg/mL) concentration of neutrophil in LDH assay (Appendix A).

### 3.2. Decondensation of Chromatin in 5 nm AgNP-Treated Neutrophils

During the process of NET formation, chromatin is decondensated to expel NETs by PAD4 or NE. To identify chromatin decondensation occurring in AgNP-treated neutrophils, human neutrophils were treated with 5 nm AgNPs for each time point (Figure 2). Confocal microscopy images revealed that the nuclei were expanded in 1–1.5 h, and NETs were observed at 3–3.5 h after treatment with 5 nm AgNPs.

### 3.3. NET Formation by 5 nm AgNPs through ROS

ROS are key factors when classifying NETs. The main sources of ROS in neutrophils are NADPH oxidase and mitochondria. In the experiment, ROS were blocked using three different ways. ROS were blocked with the ROS scavenger NAC, NADPH oxidase inhibitor (DPI), or Mitochondrial ROS inhibitor (MitoTEMPO) (Figure 3A). NAC completely reduced 5 nm AgNP-induced extracellular DNA release. However, NADPH oxidase inhibition by DPI or mitochondrial ROS inhibition by MitoTEMPO did not inhibit AgNP-induced extracellular DNA release. PMA-induced extracellular DNA release was decreased by NAC and NADPH oxidase inhibitors but not by mitochondrial ROS inhibitors. Confirmation was provided by direct measurement of ROS with flow cytometry using CM-H_2_DCFDA staining (Figure 3B). ROS production using 5 nm AgNPs was reduced only by treatment with NAC. The NADPH oxidase inhibitors or mitochondrial ROS inhibitors did not inhibit 5 nm AgNP-induced ROS production.

### 3.4. Inhibition Experiments with Non-Thiol Antioxidants in 5 nm AgNP-Treated Neutrophils

Since thiol antioxidants like NAC are reported to chelate both ROS and metal ions produced from metal NPs, non-thiol antioxidants (trolox, L-ascorbic acid) were tested (Figure 4). Human neutrophils were pretreated with trolox (Figure 4A) or L-ascorbic acid (Figure 4B) and treated with 5 nm AgNPs or PMA for 4 h. The non-thiol antioxidants inhibited some extracellular DNA release, but the inhibition was less strong than that achieved using NAC.

### 3.5. Inhibition of Phagocytosis-Lysosome Process in 5 nm AgNP-Treated Neutrophils

To determine the other ROS sources in neutrophils treated with AgNPs, phagocytosis-lysosome process was inhibited (Figure 5). The cells were treated with the inhibitors of phagocytosis (cytochalasin D, Figure 5A) or lysosome (chloroquine, Figure 5B) followed by a 4 h exposure to 5 nm AgNPs or PMA. Cytochalasin D and Chloroquine slightly inhibited NET formation induced by 5 nm AgNPs.

### 3.6. Mechanisms of NET Formation by 5 nm AgNPs

Some enzymes and granular proteins downstream of ROS are reportedly related to NET formation [32,33,34]. They include PAD, NE, and MPO. Some signaling molecules also known to induce NET formation. To investigate the mechanisms of AgNP-induced NET formation, neutrophils exposed to AgNPs were tested by inhibitors. BB-Cl-amidine (PAD inhibitor) and GW311636A (NE inhibitor) dramatically reduced extracellular DNA release in neutrophils treated with 5 nm AgNPs but ABAH (MPO inhibitor) did not (Figure 6A). BAPTA-AM (intracellular calcium chelator) also reduced extracellular DNA release in neutrophils treated with 5 nm AgNPs (Figure 6B). Signaling related inhibitors (PD98059 for MEK-1 inhibition and BML-257 for Akt inhibition) did not inhibit extracellular DNA release in neutrophils treated with 5 nm AgNPs. NETs induced by PMA were reduced by BB-Cl-amidine, GW311626A, Go6976 (PKC inhibitor), and BAPTA-AM (Figure 6A,B). Confocal image staining of DNA showed that PAD inhibitor, NE inhibitor, and intracellular calcium chelator reduced NET formation (Figure 6C).

### 3.7. Histone Citrullination and Degradation in 5 nm AgNP-Treated Neutrophils

Histone citrullination and degradation is the main mechanism of histone decondensation during NET formation [32,33,34]. PAD enzymes neutralize positively charged histone tail through citrullination [32,33]. NE cleaves histones, resulting in histone decondensation [34]. To verify the results shown in Figure 6A, histone citrullination and degradation in AgNP-treated neutrophils were evaluated (Figure 7). As expected, citrullinated histone H3 levels were gradually increased when neutrophils were treated with 5 nm AgNPs (Figure 7A). Histone H2B levels were decreased in time—and concentration—dependent manners (Figure 7B).

### 3.8. Origin of DNA in 5 nm Induced NETs 

During NET formation, the type of DNA was also determined depending on the stimulus. Following DNA isolation from neutrophils treated with 5 nm AgNPs or PMA, mitochondrial genes (*MT-ND1* and *MT-CYB*) and nuclear genes (*GAPDH* and *FAS*) were amplified from isolated DNA (Figure 8). PCR revealed that AgNP-induced NETs harbored both nuclear and mitochondrial DNA.

## 4. Discussion

NETs are double-edged swords of innate immunity [35]. They are effective tools to neutralize and combat bacteria, fungi, and some viruses. However, excessive or sterile NET formation can act as a source of autoantigen [1,2] or can promote tissue damage [3,4]. NETs are associated with several autoimmune diseases, such as SLE, rheumatoid arthritis, and vasculitis [5,10,11,12]. 

The characteristics of NETs are determined according to the stimulus. Kenny et al. showed massive results for characteristic of NETs according to five different stimuli; PMA, A23187, nigericin Candida albicans, and Group B Streptococcus [21]. They inhibited component related NETs generation such as protein kinase C, calcium, ROS, MPO, NE, CitH3, caspase. The results of study showed that each five stimuli have different mechanism for NET formation and characteristics including origin of DNA also different. The characteristics include mechanisms to expel DNA and forms and even components of NETs. Induction of NETs with *Pseudomonas aeruginosa* mucoid and non-mucoid strains revealed 50 variable protein components [13]. In addition, NETs induced by lipopolysaccharides A23187 and PMA as representative vital or suicidal NET inducers displayed differences in the release of granular enzymes, enzymes activity, and binding affinity of enzymes with NETs [14]. Therefore, it is necessary to analyze the mechanisms and characteristics of NETs depending on the stimulus.

AgNPs are one of the most commonly used NPs. They are cytotoxic to mammalian cells, mainly because of ROS generation [15]. In general, ROS generation is one of the common mechanisms in toxicity of metal NPs. Zinc oxide NPs induced ROS and lipid peroxidation in a myoblast and preadipocyte cell line, and amphipathic silica NPs induced ROS-mediated and p53-dependent apoptosis in human and rat cell lines [16,17]. ROS associated with NPs induce oxidative stress, which leads to DNA damage, unregulated cell signaling, and even cell death [36,37,38].

ROS are also important in the immune reaction of neutrophils, represented by oxidative burst [6,7,8,9]. In oxidative burst, a large amount of oxygen is consumed and superoxide is produced by NADPH oxidase. This oxidant production is important for antimicrobial defense and to regulate other immune responses of neutrophils, such as granule release and cytokine production [17]. ROS generation is also a key factor in suicidal NET formation. 

In this study, AgNPs induced the cloudy form of NETs (Figure 1) by ROS generation (Figure 3). Both are characteristics of suicidal NETs. In the ROS generation of neutrophils, most sources of ROS were NADPH oxidase-induced or of mitochondrial origin. NADPH oxidase is responsible for ROS production in most stimulated neutrophils. Chemicals such as PMA induced NADPH oxidase-dependent NET formation [22] and bacteria such as *P. aeruginosa* also induced NADPH oxidase-dependent NET formation [23]. The mitochondrial electron transport chain is another source of ROS. A23187 (calcium ionophore) induced mitochondrial ROS-dependent, NADPH oxidase-independent NET formation [24]. In contrast, polyinosinic acid (a ligand of the class A scavenger receptor) induced NET formation that is dependent on both NADPH oxidase and mitochondrial ROS [25]. However, in this study, AgNP-induced ROS in neutrophils were not dependent on NADPH or mitochondria (Figure 3). To determine the sources of ROS, neutrophils were treated with the inhibitors of lysosome and phagocytosis (Figure 5). Phagocytosis and lysosome inhibitor slightly reduced extracellular DNA release. The reduction of NET formation by the inhibitor for phagocytosis indirectly indicated that AgNPs were in part phagocytosed by neutrophils (Figure 5A). This result was consistent with that of lysosome inhibition (Figure 5B). The rest of infiltrated AgNPs might have been penetrated into the cells by diffusion. Similarly, lysosomal damage and NET formation in neutrophils were reported for diamond NPs [19]. NPs in phagolysosomes can release ion in an acidic environment, which can damage lysosomal membrane and induce ROS. This mechanism is called the “lysosome-enhanced Trojan horse effect” [26]. ROS generation by NPs can involve membrane damage, lysosomal damage, and ionized NPs. Since NAC is a thiol antioxidant and can chelate metal ions, the effect of non-thiol antioxidants was examined (Figure 4). Trolox and L-ascorbic acid reduced extracellular DNA but not as dramatically as NAC. These results indicated that AgNP-induced ROS generation might be related to ionized silver. The release of silver ions by AgNPs is reportedly one of the cytotoxicity mechanisms of AgNPs. The toxicity of AgNPs in *Caenorhabditis elegans* is dependent on dissolved silver [27]. The ROS-dependent cytotoxicity of AgNPs in human hepatoma cells was dependent on silver ion [28]. 

AgNP-induced NETs were PAD4- and NE-dependent (Figure 6A). These results indicate that histone citrullination by PAD4 and histone degradation by NE occurred during NET formation by AgNPs. The citrullination and degradation of histones were observed using 5 nm AgNPs (Figure 7). The chelator of intracellular calcium, which is a cofactor of PAD4, also reduced NET formation (Figure 6B). However, other signaling inhibitors did not show significant results. 

Neutrophils use their nuclear DNA as the backbone of NETs. However, in some cases, they use their mitochondrial DNA, and the choice of DNA type depends on the stimulus [29,39]. Low-density granulocytes, which are a subtype of neutrophils, release mitochondrial DNA in patients with SLE in a mitochondrial-superoxide-dependent manner [29]. Additionally, lipopolysaccharide induces the release of mitochondrial DNA in granulocyte-macrophage colony-stimulating factor-primed human neutrophils [39]. In AgNP-induced NETs, both nuclear and mitochondrial DNA were observed (Figure 8). DETA-NONOate and PMA treatment in neutrophil show similar PCR results [40]. According to Lood et al., ribonucleoprotein-containing immune complex treatment and DNA oxidation induced NETs and increased ratio of 16S:18S DNA in NETs suggesting mitochondrial DNA and chromosomal DNA can be coexisting in NETs and that ratio can be change depending on stimuli [29]. 

## 5. Conclusions

Neutrophils induced NET formation in a 5 nm AgNPs treatment. Chromatin decondensation and extracellular DNA release was observed. AgNP-induced ROS generation was decreased by ROS scavengers, such as NAC, trolox, and L-ascorbic acid. However, inhibition of NADPH oxidase and mitochondrial ROS, which are representative ROS sources of neutrophils, did not reduce ROS or extracellular DNA release. In the inhibitor study, AgNP-induced NETs were only decreased by inhibitors for PAD4, intracellular calcium, and NE. The intracellular calcium chelator also reduced NET formation because calcium is a cofactor of PAD4. The results of phagocytosis inhibition indirectly showed that 5 nm AgNPs entered neutrophils partly by phagocytosis. Chloroquine, a lysosome inhibitor, also reduced extracellular DNA release in 5 nm AgNP-treated neutrophils. Other inhibitors related to cell death and signaling inhibitors did not reduce extracellular DNA release. The PCR results show that, the AgNP-induced NETs consisted of both nuclear and mitochondrial DNA. In this study, we investigate mechanism of AgNP-induced NETs and suggest its effective inhibitors. Neutrophils are the most abundant leukocytes and the first cells encountered foreign antigen. These results may contribute to the toxicity study and risk assessment of AgNPs exposure to human.

## Figures and Tables

**Figure 1 biomolecules-11-00317-f001:**
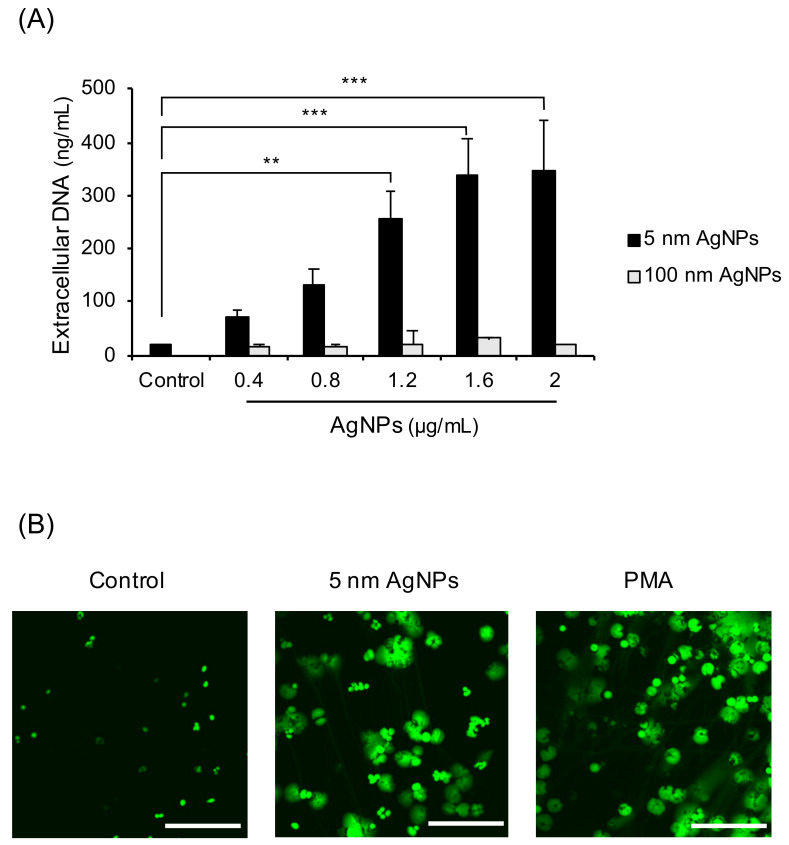
Induction of NET formation by 5 nm AgNPs. (**A**) Human neutrophils were treated with AgNPs for 4 h, and extracellular DNA was measured. (**B**) Neutrophils were treated with 1.6 μg/mL of 5 nm AgNPs or 5 ng/mL PMA for 4 h, and the DNA was stained using SYTOX Green. Scale bar: 100 µm. One-way ANOVA was used to determine the significance (** *p* < 0.01, *** *p* < 0.001). All experiments were performed in duplicate and repeated three times with different volunteers.

**Figure 2 biomolecules-11-00317-f002:**
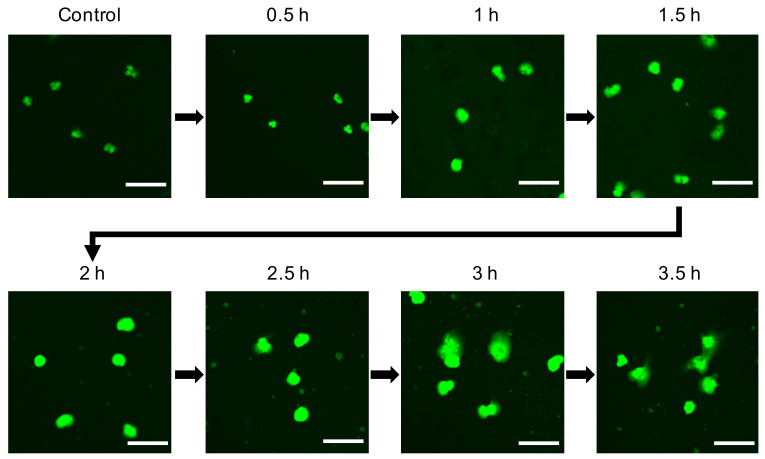
Decondensation of chromatin in 5 nm AgNP-treated neutrophils. Human neutrophils were treated with 1.6 μg/mL 5 nm AgNPs on poly-l-lysine-coated chamber slides for each indicated time point and fixed with 4% paraformaldehyde. The DNA of cells was stained with SYTOX Green. Scale bar: 20 µm. This experiment was performed in duplicate and repeated three times with different volunteers.

**Figure 3 biomolecules-11-00317-f003:**
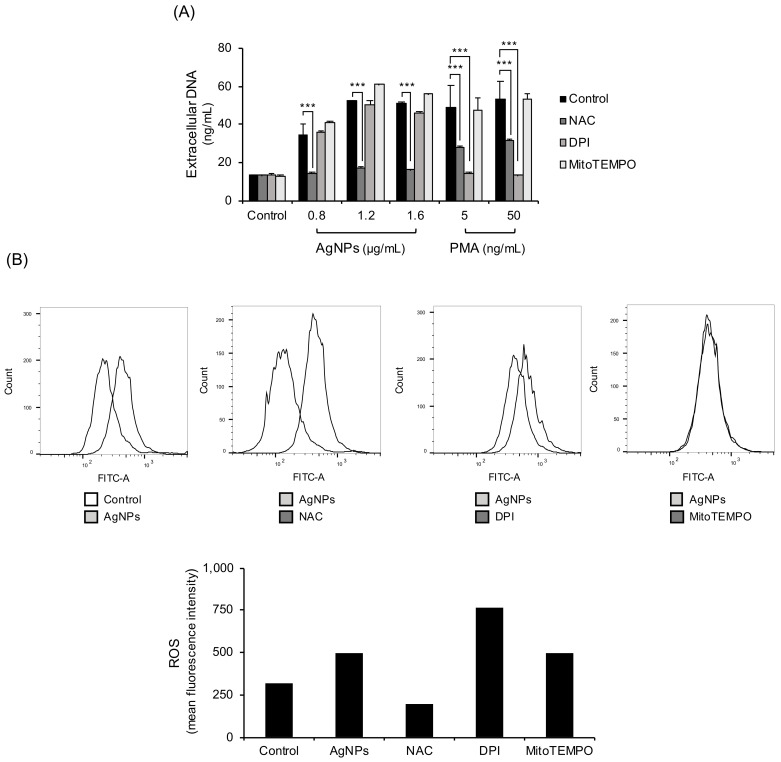
NET formation by 5 nm AgNPs through ROS. (**A**) Human neutrophils were pretreated with ROS inhibitors, including NAC (10 mM, 1 h) (ROS scavenger), DPI (10 μM, 30 min) (NADPH oxidase inhibitors), MitoTEMPO (10 μM, 1 h) (mitochondria-targeted ROS inhibitors). The samples were treated with 5 nm AgNPs or PMA for 4 h, and extracellular DNA was measured using a fluorescence microplate reader. (**B**) Human neutrophils were pretreated with CM-H_2_DCFDA and ROS inhibitors. The cells were incubated with 1.6 μg/mL 5 nm AgNPs or 5 ng/mL PMA for 1.5 h and analyzed by flow cytometry. Two-way ANOVA was used to determine the significance (*** *p* < 0.001). All experiments were performed in duplicate and repeated three times with different volunteers.

**Figure 4 biomolecules-11-00317-f004:**
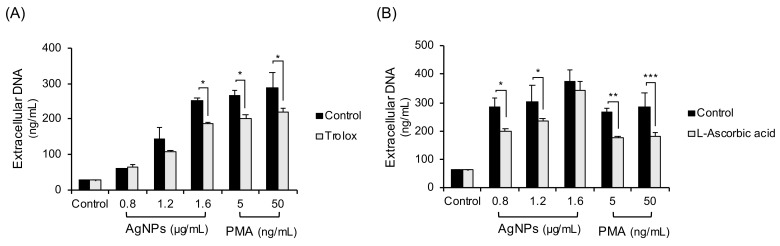
Inhibition experiments with non-thiol antioxidants in 5 nm AgNP-treated neutrophils. Human neutrophils were pretreated with non-thiol antioxidants: (**A**) trolox (100 μM, 1 h) or (**B**) L-ascorbic acid (20 μM, 1 h) and treated with AgNPs or PMA for 4 h. Extracellular DNA were measured using a fluorescence plate reader. Two-way ANOVA was used to determine the significance (* *p* < 0.05, ** *p* < 0.01, *** *p* < 0.001). All experiments were performed in duplicate and repeated three times with different volunteers.

**Figure 5 biomolecules-11-00317-f005:**
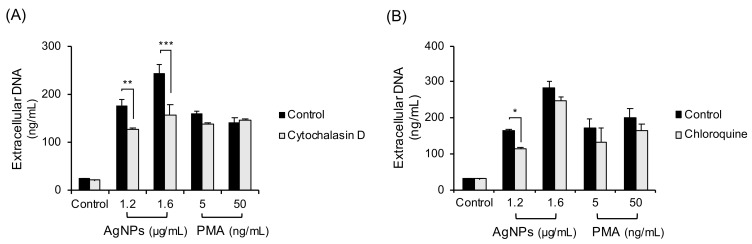
Inhibition of phagocytosis-lysosome process in 5 nm AgNP-treated neutrophils. Human neutrophils were pretreated with inhibitors that included (**A**) cytochalasin D (20 μM, 20 min) (phagocytosis inhibitor) or (**B**) chloroquine (3 μM, 1 h) (lysosome inhibitor) and followed by 4 h exposure to AgNPs or PMA. Extracellular DNA were measured by a fluorescence plate reader. Two-way ANOVA was used to determine the significance (* *p* < 0.05, ** *p* < 0.01, *** *p* < 0.001). All experiments were performed in duplicate and repeated three times with different volunteers.

**Figure 6 biomolecules-11-00317-f006:**
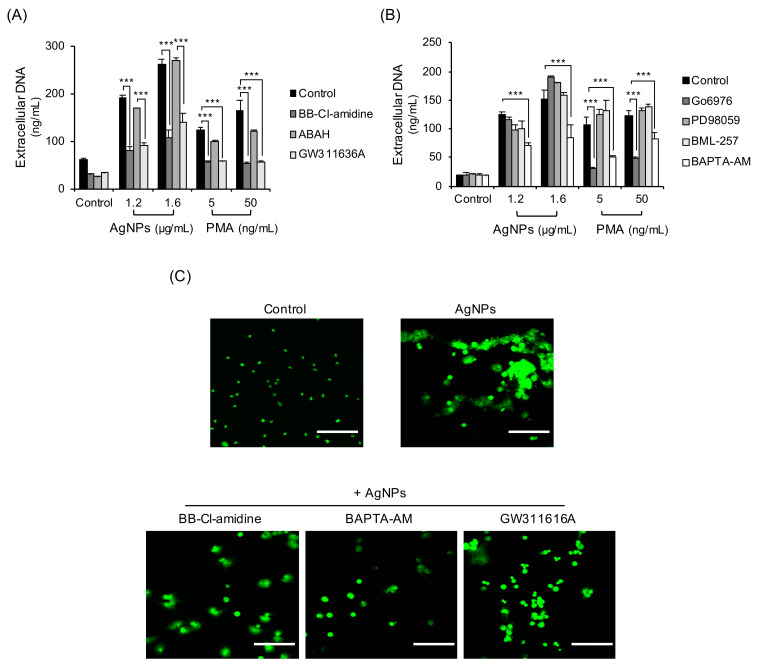
Mechanisms of NET formation by 5 nm AgNPs. Human neutrophils were pretreated with inhibitors that included (**A**) BB-Cl-amidine (8.8 μM, 20 min) (PAD inhibitor), ABAH (500 μM, 30 min) (MPO inhibitor), GW311616A (20 μM, 30 min) (neutrophil elastase inhibitor), (**B**) Go6976 (1 μM, 30 min) (PKC inhibitor), PD-98059 (40 μM, 20 min) (MEK-1 inhibitor), BML-257 (20 μM, 20 min) (Akt inhibitor), and BAPTA-AM (10 μM, 15 min) (intracellular calcium chelator). This was followed by 4 h exposure to 5 nm AgNPs or PMA. Extracellular DNA was measured by a fluorescence plate reader. (**C**) Human neutrophils were pretreated with the inhibitors BB-Cl-amidine, BAPTA-AM, and GW311616A and treated with 1.6 μg/mL 5 nm AgNPs for 4 h. The cells were fixed with 5% paraformaldehyde and stained with SYTOX Green for DNA staining. Scale bar: 100 µm. Two-way ANOVA was used to determine the significance (*** *p* < 0.001). All experiments were performed in duplicate and repeated three times with different volunteers.

**Figure 7 biomolecules-11-00317-f007:**
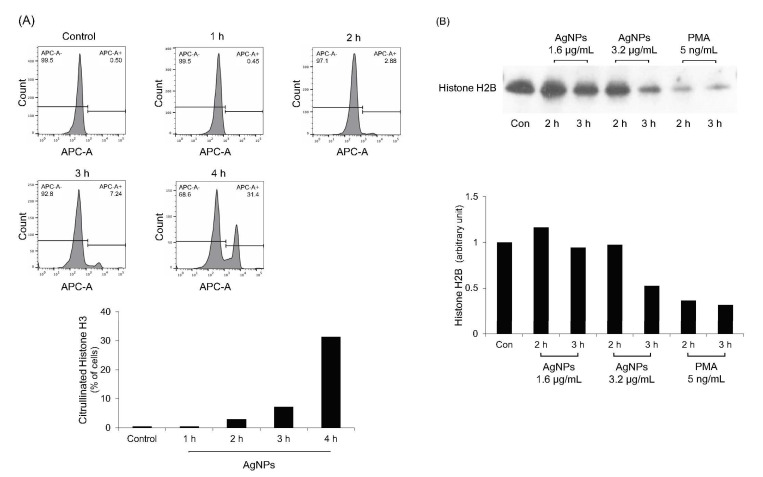
Histone citrullination and degradation in 5 nm AgNP-treated neutrophils. (**A**) Human neutrophils were treated with 1.6 μg/mL 5 nm AgNPs for each indicated time point and citrullinated histone H3 was measured using flow cytometry. (**B**) Western blotting of histone H2B. Human neutrophils were treated with 5 nm AgNPs or PMA for 2 or 3 h, and total histones were isolated. Histone H2B levels were detected by western blotting to verify histone degradation. All experiments were performed in duplicate and repeated three times with different volunteers.

**Figure 8 biomolecules-11-00317-f008:**
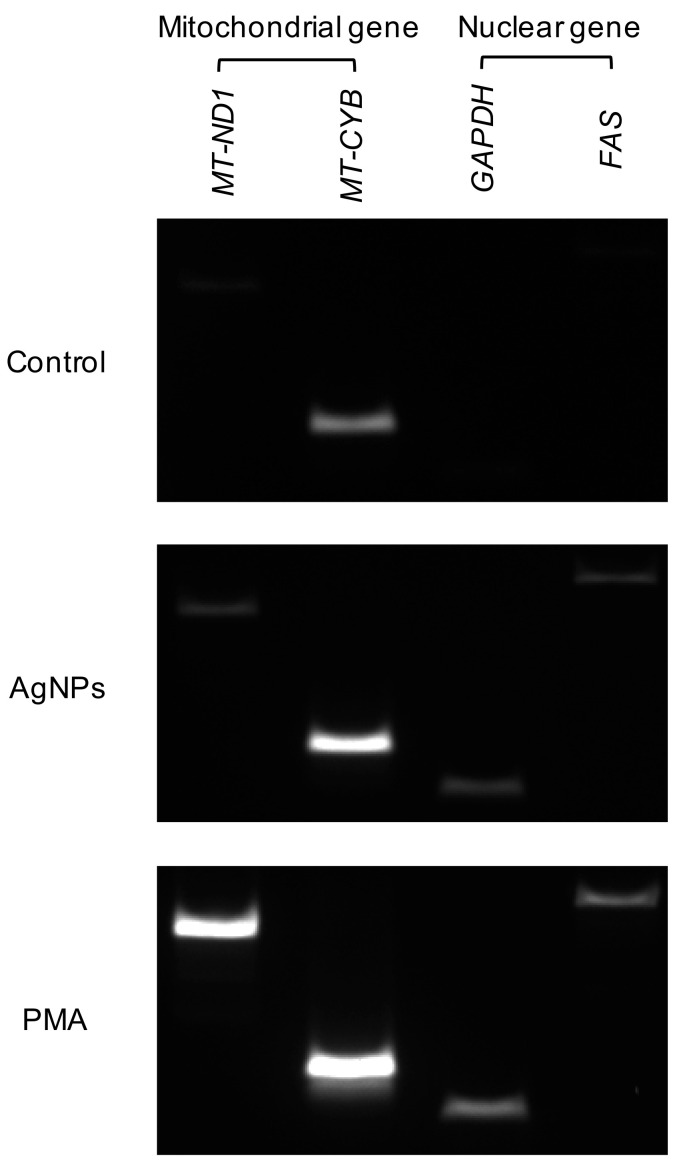
Origin of DNA in 5 nm AgNP-induced NETs. Human neutrophils were treated with 1.6 μg/mL 5 nm AgNPs or 5 ng/mL PMA for 4 h, and DNA were isolated from the supernatant. Mitochondrial genes (*MT-ND1* and *MT-CYB*) and nuclear genes (*GAPDH* and *FAS*) were amplified from isolated DNA. This experiment was performed in duplicate and repeated three times with different volunteers.

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
