# Peer review of "Silver Nanoparticles Induce Neutrophil Extracellular Traps Via Activation of PAD and Neutrophil Elastase"

_biomolecules, 2021, doi:10.3390/biom11020317_

Round 1

Reviewer 1 Report

The overall idea of the manuscript is valid. The reason behind the work makes sense and the manuscript is well structured. And despite the scientifically sound work, this topic is not new. Therefore, the references are recent, but poor, as the novelty of this work is not high. There are several works already published (e.g. DOIs: 10.1155/2019/3560180; 10.1016/j.intimp.2015.06.030; 10.1016/j.taap.2008.09.015). The English language are generally correct but needs to be fully revised. Authors may take in account the following comments and suggestions, in order to improve their manuscript and enable it to be considered for publication:

- Introduction section should be improved with a clear explanation on the innovative nature of the work, as other works on this theme already exist, as previously pointed out (see the DOIs above, as example);

- In the Introduction section other NPs than AgNPs are referred. Why? Improve it;

- In the Introduction section remove the last sentence;

- The Methods texts do not have references. Why? All the methods are original? Improve it;

- Refer how many assays were done for each method. And with how many volunteers?

- In the Methods section refer the concentrations used of each reagent or NPs tested;

- In the ROS production method, refer how was done the seeding procedure of neutrophils. Also explain why the cells were not washed prior to cytometry;

- In the results section, the number of assays done (n) are missing; add it; several graphs do not have error bars, add them;

- In figure 3B; “mean intensity” refers to what? Improve the yy axis designation; add the bar of PMA alone; and explain why DPI did not diminished the signal;

- In figure 3, what are the graphs with the count; they are not mentioned in the figure caption; correct it;

- The explanation about the selected incubation times for each assays should be added;

- Based in figure 5, the conclusion “Cytochalasin D and Chloroquine slightly inhibited NET formation.” does not seem totally correct; complete it;

- A reference should be made to the concentrations of AgNPs tested; are they physiologically meaningful? Add this justification to the manuscript;

- Figures should not be present in the Discussion section;

- The Discussion section should be improved according to the already published works in the literature, by comparing the obtained results with the ones already published; and if they are not comparable, explain why is that.

Reviewer 2 Report

Kang et al. present an interesting study investigating the impact of AgNPs on human neutrophils, and NETs.

This can be of interest, due to the use of silver in cosmetics and biomedical application.

The study is well conducted, with nice findings. Some minor concerns should be addressed however.

Introduction: What is meant by the chance to expose to AgNPs are increasing? End of first paragraph in introduction, ROS are key factors...please give reference. Third paragraph, NETs induced by various stimuli, please read and maybe cite Kenny et al. (https://doi.org/10.7554/eLife.24437.001) and Branzk et al. (https://doi.org/10.1038/ni.2987) here. 

Material and Methods: 2.1. please give here final concentrations of used reagents. 2.2. Polymyxin B induces NETs (doi:10.1042/BJ20140778), has this been considered when in bringing in contact with PMNs? 2.3 was purity of cells determined? Why were different time points used for ROS measurement and NET assessment? 2.7. Why again different time points? 2.8 How long were cells stimulated before DNA purification? 

Results: 3.1. Can the differences in effect by different-sized NPs be due to differences in origin and preparation? Fig. 1A, how can this be explained in context of findings by Branzk et al.? Figure 2, add arrows to indicate NET structures for better understanding. Fig. 3A why 2 different concentration of PMA, and why these concentrations? Fig. 3B, Why is ROS higher in DPI treatment? Please explain. Fig.3 please give figure legend for flow panels, it is not easy to understand otherwise.  3.6 Ref is missing for first sentence (some enzymes and granular proteins....). 3.7. References missing for first 3 sentences! Fig 7B, why not similar time points like Fig. 7A? Fig.7 what are ACP positive cells (%)? Figure legend missing for this graph? 

How is LDH explained? why only for smaller NPs? Was necrosis excluded? 

Discussion: Reference needed for double-edged sword (10.4049/jimmunol.1201719). Second sentence, please change "destroy" for "combat". Second paragraph: Discuss findings by Kenny et al. and Branzk et al.  Third paragraph: Reference missing for ROS association to cell death; similar 4. paragraph, ROS key factor for suicidal NETs, add reference! How do the authors come to conclusion that AgNPs were phagocytosed? Where are the according results? Reference missing for mitochondrial DNA release depending on stimulus. Please discuss more that both nuclear and mitochondrial DNA was found in response to AgNPs, has this been described by others, etc.?

Conclusion: How is phagocytosis of NPs shown? What is the study of AgNPs exposure to human? please explain what is meant by this.
